# Effects of Kefir Consumption on Gut Microbiota and Athletic Performance in Professional Female Soccer Players: A Randomized Controlled Trial

**DOI:** 10.3390/nu17030512

**Published:** 2025-01-30

**Authors:** Ece Öneş, Mutlucan Zavotçu, Nida Nisan, Murat Baş, Duygu Sağlam

**Affiliations:** 1Department of Nutrition and Dietetics, Institute of Health Sciences, Acibadem Mehmet Ali Aydinlar University, Istanbul 34752, Türkiye; 2Movement and Training Sciences Program, Institute of Health Sciences, Marmara University, Istanbul 34865, Türkiye; mutlucanzavotcu@gmail.com; 3Fatih Vatan Sports Club, Istanbul 34091, Türkiye; nidaniisan@gmail.com; 4Department of Nutrition and Dietetics, Faculty of Health Sciences, Acibadem Mehmet Ali Aydinlar University, Istanbul 34752, Türkiye; murat.bas@acibadem.edu.tr (M.B.); duygu.saglam@acibadem.edu.tr (D.S.)

**Keywords:** kefir, gut microbiota, short-chain fatty acids, athletic performance, professional female soccer players, functional foods, microbiota diversity, VO_2_max, sports nutrition

## Abstract

Background/Objectives: This study aimed to determine the impact of the daily consumption of kefir on the gut microbiome, body composition, and athletic performance of professional female soccer players. Methods: The participants encompassed 21 females aged 18–29 years who were assigned to one of the two groups: the experimental group, which comprised females who consumed 200 mL of kefir daily for 28 days, and the control group, which comprised females who continued with their normal diet. Anthropometric measurements, dietary intake, the composition of the gut microbiome through 16S rRNA gene sequencing, and an athletic performance test known as the 30-15 Intermittent Fitness Test were performed before and after the intervention. Results: The results of this study revealed that the consumption of kefir increased the microbial diversity (Shannon and Chao1 indices), wherein a significant increase was noted in the abundance of *Akkermansia muciniphila* and *Faecalibacterium prausnitzii*, microorganisms that regulate energy metabolism and have anti-inflammatory effects. Furthermore, the athletic performance variables, including VO_2_max (mL.kg^−1^.min^−1^) and finishing speed (km/h), were strongly related to the abundance of these short-chain fatty acid-producing bacteria. A link between the microbiota profile and the dietary intake of fiber and protein as well as the body composition measurements was also established. Conclusions: This study indicated that kefir consumption can positively affect the gut microbiota, which could in turn affect the athletes’ performance. Therefore, to determine the effects of kefir as a functional food in sports nutrition over a longer period, more research should be conducted.

## 1. Introduction

Recently, the expanding sports industry has sparked increased interest in tailored strategies aimed at enhancing sports performance. Such strategies include training practices, alternative recovery techniques, and nutritional strategies [1]. In the context of all these developments, gut microbiota-targeted dietary interventions have been evaluated for enhancing athletic performance. The inclusion of dietary probiotics, which can modulate the gut microbiota, is suggested as a feasible and effective approach for boosting athletic performance [2].

Kefir, a type of fermented milk product, has a high probiotic value, resulting in favorable health effects. Kefir composition changes depending on the type of grain used, type of milk, and fermentation and storage conditions [3]. It frequently includes *Lactobacillus*; *Lactococcus*; *Streptococcus leuconostoc*; acetic acid bacteria; and yeasts including *Saccharomyces*, *Kluyveromyces*, and *Candida* [4]. Moreover, kefir has been shown to decrease serum cholesterol level and inflammation, possess anticarcinogenic properties, and improve gastrointestinal health [5]. Moreover, kefir consumption stimulates the gut microbiota to produce beneficial metabolites, including short-chain fatty acids (SCFAs) [6]. SCFAs can efficiently facilitate increased glycogenesis by improving lipid absorption and oxidation [7]. In an animal study, adding kefir to their diet for 28 days inhibited gut microflora alterations; decreased plasma creatine kinase, ammonia, and lactate levels; and promoted fatigue resistance by increasing swimming time to exhaustion and forelimb grip strength [8]. A previous human study performed a 28-day kefir supplementation and reported no significant changes in the composition of the gut microbiota; instead, it showed that blood lactate levels are decreased and endurance performance increased [9].

Despite several potential benefits, the possible effects of regular kefir consumption on gut microbiota modulation and athletic performance have not been adequately investigated in professional athletes. This study aimed to fill this gap in the scientific literature by examining the effects of regular kefir consumption on gut microbiota and athletic performance in professional female soccer players and add valuable insights into the use of fermented foods in sports nutrition. It was hypothesized that regular kefir consumption would enhance gut microbiota diversity and composition, as well as improve athletic performance metrics such as VO_2_max and finishing speed.

## 2. Materials and Methods

### 2.1. Study Design

This study was designed as a randomized controlled trial to investigate the effects of regular kefir consumption on the gut microbiota and athletic performance of professional female soccer players. The sample size was determined using a power analysis conducted with the GPower software (Version 3.1.9.7). A statistical power of 80%, a significance level (α) of 0.05, and an expected effect size (d) of 0.8 were used to calculate the minimum required sample size. Based on this analysis, a total of 24 participants were required for a parallel design study, evenly distributed between the experimental group (kefir consumption, 12 participants) and the control group (no intervention, 12 participants), ensuring a 1:1 allocation ratio. The random allocation sequence was generated using a stratified randomization approach based on age categories to ensure balanced distribution across the experimental and control groups. Specifically, participants were first categorized into age strata, and randomization was performed independently within each age stratum using a computer-generated random number list. This approach minimized the potential confounding effects of age on outcomes. Blinding was not feasible in this study due to the distinct taste and texture of kefir, which could not be adequately masked by alternative placebo options such as milk or ayran (fluid form of yogurt). The kefir intervention lasted 28 days, and the soccer players were randomized into two groups before the intervention began. The case group consumed 200 mL/day of kefir for 28 days, whereas the control group did not receive any intervention. This study was conducted during the season in one of the Turkish Women’s Soccer Super League teams. The training programs underwent no intervention. To rule out different training loads, soccer players from the same team were included in this study. This study was conducted in accordance with the principles of the Declaration of Helsinki and was medically ethical according to the Acıbadem University and Acıbadem Healthcare Institutions Medical Research Ethics Committee (decision number: 2024-4/148). This study was also registered with ClinicalTrials.gov (registration number: NCT06753422).

### 2.2. Participants and Kefir Supplementation

At the beginning of this study, 25 professional female soccer players were included. However, four participants were excluded owing to inconclusive results of repeated DNA sequencing. The flow of participants throughout the study, including allocation, follow-up, and analysis, is shown in Figure 1. Therefore, this study was conducted with 21 professional female soccer players aged 18–29 years who play on the same team in the Turkish Women’s Soccer Super League. The recruitment period spanned from 1 May 2024 to 7 May 2024, while the follow-up period extended from 8 May 2024 to 5 June 2024. All soccer players who volunteered to participate met the inclusion criteria and did not meet any of the exclusion criteria of this study. The following were the exclusion criteria: having used antibiotics in the last month and/or currently using antibiotics; having used probiotic, prebiotic, or symbiotic supplements in the last 3 months; with milk protein allergy or lactose intolerance; having a chronic disease and/or being on long-term medication; and failed to comply with regular kefir consumption.

Participants in the experimental group consumed 200 mL of kefir daily as a recovery drink after training/a match or at any time on rest days for 28 days. Under cold chain conditions, the research dietician delivered the kefir drinks to the soccer club. Before study initiation, the soccer players were trained on cold chain conditions. Kefir daily consumption was performed at the soccer club, and the research team performed compliance checks. Soccer players in the control group continued their daily diet routine. In this study, a single kefir brand was consistently used throughout the intervention period to ensure standardization and minimize variability in product composition, probiotic content, and nutritional profile. This approach reduces potential confounding effects arising from differences in kefir brands and supports the reliability of the intervention outcomes. Additionally, the kefir used in the study was sourced from a commercially available, standardized product with consistent fermentation processes, bacterial strains, and quality control measures, ensuring reproducibility and reliability of the intervention. The kefir provided during the intervention contained 140 kcal, 7.3 g of fat, 4.2 g of carbohydrates, and 9.2 g of protein per 250 mL, ensuring a balanced nutritional profile that aligns with the needs of athletes.

### 2.3. Anthropometry and Body Composition

All soccer players (*n* = 21) underwent anthropometry and body composition analyses twice, at the beginning and end of this study. Height measurements were performed using the SECA 213 Stadiometer, barefoot, feet together from the heels, in an upright position, and in the Frankfurt plane. Body weight, fat mass, and fat-free mass were measured using the BIA method using the Tanita BC 545-N (Tanita Corporation, Tokyo, Japan). Regarding body fat percentage, skinfold thickness measurements were taken from seven regions (triceps, subscapular, biceps, suprailiac, abdomen, thigh, and calf) using the Holtain Skinfold caliper; regarding body density, the Jackson–Pollock equation was employed for females, and the Siri equation was subsequently used for calculating the body fat percentage. All measurements were performed in the morning following an overnight fast and wearing light training clothes.

### 2.4. Dietary Assessment

Nutritional status was analyzed twice, at the beginning and end of the study, using 3-day food consumption records from all soccer players (*n* = 21). Soccer players were asked to fill in their food consumption records for a single training day, a double training day, and an off day. Before study initiation, to minimize the possibility of keeping incorrect and/or incomplete records, the researcher dietician provided nutrition education to the soccer players on keeping accurate food records. Following completion, the Nutrition Information System (BEBIS) 7.2 software was used for analyzing all records.

### 2.5. Athletic Performance Assessment

All participants underwent the 30-15 Intermittent Fitness Test (IFT) twice, at the beginning and end of the study, by the team head coach, following a standard protocol. According to the results of a recent systematic review, the 30-15 IFT has excellent test–retest reliability for both maximal velocity and peak heart rate. It has been identified as a reliable measure of fitness for use in both research and sports practice [10]. In addition, a study by Čović et al. demonstrated that the 30-15 IFT is also highly reliable for female soccer players, effectively detecting individual performance changes [11].

The 30-15 IFT measures the maximal aerobic speed and maximal oxygen consumption (VO_2_max) of soccer players. During the test, soccer players ran for 30 s at the specified speed (km/h) followed by a 15 s rest. The running speed starts at 8 km/h and increases by 0.5 km/h every 30 s.

### 2.6. DNA Extraction from Stool Samples (DiaRex)

Stool samples were processed using the DiaRex^®^ Stool Genomic DNA Extraction Kit (Cat No: SD-0323, Diagen, Ankara, Türkiye). Briefly, 250 µL of the stool lysis (SLD) solution was added to 25–50 mg of the stool sample. Subsequently, 15 mg of glass beads and 10 zirconium beads were added, and the mixture was homogenized using a homogenizer (FP120; Thermo Fisher Scientific, Waltham, MA, USA) at 4000 rpm for 2 × 20 s.

Following homogenization, 25 µL of the proteinase K (PKD) solution was added, and the mixture was incubated at 56 °C for 60 min. Following incubation, the entire content was centrifuged at 5000× *g* for 5 min, and the supernatant was transferred to a new tube.

For the supernatant, 200 µL of the lysis (LBD) solution was added, followed by incubation at 70 °C for 10 min. Following incubation, 250 µL of absolute ethanol was added to the lysate, and the entire mixture was transferred to the column. The column was centrifuged at 8000× *g* for 1 min, and the flow-through was discarded.

Subsequent washing steps were performed according to the kit protocol. Finally, 100 µL of the elution (EBD) solution was added to the column, incubated for 2 min, and centrifuged at 8000× *g* for 1 min to elute the genomic DNA.

### 2.7. 16S Amplicon Sequencing

The extracted DNA was amplified using 16S V3-V4 primer sets, and library preparation was handled using the Nextera XT DNA library preparation kit and indexes (Illumina, San Diego, CA, USA). Pooled libraries cleaned-up with specific size selection were applied following the manufacturer’s protocol (AMPure XP, Beckman Coulter, Brea, CA, USA). Following library preparation, the MiSeq (Illumina) instrument was used to run sequencing.

### 2.8. Bioinformatics Analysis

Pair-end Illumina reads (2 × 250) were imported to the Qiime2 environment [12]. Quality clipping, chimera detection, and cleaning of reads implemented through the Qiime2 Dada2 pipeline (via q2-dada2) [13]. Bases that had a low phred score (<Q30) were cut out. Amplicon sequence variants generated by Dada2 were mapped to the database [14,15]. The Phyloseq (30) object was developed from the Qiime2 artifact files in the R 4.1 environment [16,17]. Alpha diversity assessment, which was used for evaluating the diversity of related taxonomic units in a sample, was interpreted using three different indices, including Chao1, Shannon, and Simpson. *p* values between groups were calculated using the Kruskal–Wallis test [18]. Beta diversity analysis, used for assessing taxonomic differences between individuals, was calculated on the basis of jaccard, Bray–Curtis, weighted, and unweighted unifrac. Microbial compositional differences between the groups were analyzed using a permutational multivariate analysis of variance (PERMANOVA). This non-parametric test was applied to assess differences based on Bray–Curtis and Jaccard distance metrics, with statistical significance determined using 999 permutations. Additionally, the Multi-Response Permutation Procedure (MRPP) was conducted as a complementary non-parametric method to evaluate the overall compositional differences between groups. Both the Bray–Curtis and Jaccard distance metrics were applied in the MRPP analysis to assess within-group homogeneity and between-group differences. Specific differences between the groups were determined using differential abundance analysis, Deseq2 R pack [19]. To show statistically significant taxonomies, a linear discriminant analysis effect size analysis was performed between the groups [20]. A Spearman correlation analysis was used to assess the relationships between microbial species, dietary intake, and performance measures. Correlation coefficients (r) were interpreted as follows: 0.1–0.3 as weak, 0.3–0.5 as moderate, and >0.5 as strong associations, with statistical significance set at *p* < 0.05.

## 3. Results

### 3.1. Baseline Characteristics

The baseline characteristics of the experimental and control groups were mostly similar as shown in Table 1. The experimental group (24.42 ± 2.52 years) had a slightly greater mean age than the control group (22.14 ± 3.61 years). Only minor differences in body weight, fat mass, and body fat percentage were observed between the groups; the experimental group had a mean body weight of 60.32 ± 5.44 kg and a body fat percentage of 17.02% ± 2.49%, whereas the control group had a mean body weight of 58.03 ± 7.64 kg and a body fat percentage of 16.43% ± 2.80%. The experimental group had a slightly greater fat-free mass (48.16 ± 3.58 kg) than the control group (46.88 ± 5.49 kg). A statistical analysis confirmed no significant differences between the groups at baseline (*p* > 0.05 for all comparisons).

### 3.2. Effects of Regular Kefir Consumption on Microbial Diversity and Composition

It is believed that kefir consumption can positively impact the diversity and structure of the gut microbiota. In this section, the changes in the alpha and beta diversity of the microbial community following kefir consumption are described. Alpha diversity measures, including the Shannon, Chao1, and Simpson indices, were used for quantifying the microbial richness and evenness of a sample, that is, the number of different microbial species and their distribution. Beta diversity, visualized through principal coordinate analysis (PCoA) based on Bray–Curtis distances, showed variation in the microbial composition between samples. These results help determine how kefir can influence the microbial community structure in a more detailed manner at the individual and group levels.

#### 3.2.1. Alpha Diversity

The changes in microbial diversity and richness were evaluated with Shannon diversity and Chao1 richness indices when kefir consumption was considered. The present study showed that the E_post group had the highest microbial diversity and richness compared with the other groups, and the C_pre group had the lowest values. This finding is further depicted in Figure 2.

In this study, the Shannon diversity index was employed for assessing the microbial alpha diversity of experimental (E_post, E_pre) and control (C_post, C_pre) samples. The findings indicate that the E_post and E_pre groups had higher microbial diversity than the control groups, and the E_post group had the highest median diversity value. The E_post group had a slightly higher Shannon index than the E_pre group; however, the difference was not statistically significant (*p* > 0.05). The C_pre group exhibited a lower microbial diversity than the E_post group, and this difference was also significant (*p* < 0.05). Moreover, according to the Simpson index, the E_post group exhibited higher microbial diversity compared to the C_pre group, with a statistically significant difference (*p* = 0.0238). These results are detailed in the Appendix A.

To assess the microbial richness, the Chao1 richness index was employed by calculating the overall number of species that were present in each community with particular focus on the hard-to-detection species. The findings in the Figure 2 show that the post-experimental group (E_post) had the highest microbial diversity, as indicated by the highest median Chao1 score. Conversely, the control pre-experimental group (C_pre) had the lowest diversity and a narrower interquartile range. The Chao1 richness index for E_pre was lower than that for E_post but remained higher than that for both control groups (C_post and C_pre). However, when compared using statistical tests, no difference was noted between E_post and E_pre (*p* > 0.05).

#### 3.2.2. Beta Diversity

The Bray–Curtis distance metric was used for determining the compositional distance between the samples from the experimental (E_post, E_pre) and control (C_post, C_pre) groups, and the results were represented by PCoA. The two groups of experimental (E_post, E_pre) were more closely related than those of the two control groups (C_post, C_pre). Conversely, the C_pre group was the most dispersed.

As shown in the Figure 3 and Appendix A, a statistical analysis using a permutational multivariate analysis of variance (PERMANOVA) revealed significant differences in microbial composition between the experimental and control groups (Bray–Curtis, *p* = 0.018; Jaccard, *p* = 0.016). However, within the experimental groups, the differences between pre- and post-intervention (E_post vs. E_pre) were not statistically significant (*p* > 0.05).

An MRPP analysis was conducted alongside PERMANOVA to further assess microbial compositional differences. The MRPP analysis results align with the PERMANOVA findings, indicating significant differences in microbial community structures between the experimental and control groups. Detailed MRPP and PERMANOVA results, including distance metrics and significance values, are provided in the Appendix A.

### 3.3. Changes in Genus, Phylum, and Species Levels Associated with Regular Kefir Consumption

It has been demonstrated that kefir consumption can modify the gut microbiota structure at different taxonomic levels. Differences in the microbial diversity occurred at the phylum, genus, and species levels with alterations in the ratio of certain microbial populations. At the phylum level, the ratio of the *Firmicutes* to the *Bacteroidetes* was not altered, whereas the *Verrucomicrobia* and the *Euryarchaeota* were higher in the experimental group. At the genus and species levels, the microbial populations, including *Akkermansia muciniphila* (*A. muciniphila*) and *Prevotella copri* (*P. copri*), were differentially affected.

#### 3.3.1. Phylum Level Taxonomic Changes

Figure 4 shows that the phyla *Firmicutes* and *Bacteroidetes* are the dominant phyla in all the conditions and have contributed to most of the gut microbiome. In the experimental groups, some potentially pathogenic *Proteobacteria* (E_post, 2.4%; E_pre, 2.6%) were reduced, whereas increases in *Verrucomicrobia* (E_post, 3.3%; E_pre, 1.9%) and *Euryarchaeota* (E_post, 3.0%; E_pre, 1.4%) were observed.

#### 3.3.2. Genus- and Species-Level Changes

The genus-level relative abundance analysis (Figure 5) showed the dynamics of the gut microbiota and the possible kefir consumption-induced changes. *Prevotella*, *Bacteroides*, and *Faecalibacterium* were the dominant genera in all the groups. Kefir consumption reduced the relative abundances of some of the dominant genera, including *Bacteroides* (E_pre, 15.7%; E_post, 13.5%) and *Faecalibacterium* (E_pre, 14.8%; E_post, 13.6%), and enhanced the relative abundances of potentially healthy bacteria, including *Akkermansia* (E_pre, 2.4%; E_post: 4.1%) and *Bifidobacterium* (E_pre, 2.4%; E_post, 4.1%).

At the species level (Appendix A), specific bacterial shifts further highlighted the impact of kefir. *A. muciniphila*, a key player in gut mucosal integrity, significantly increased following kefir consumption (E_post, 8.5%; E_pre, 4.1%). The species *Roseburia faecis* (*R. faecis*), known for its role in SCFA production, showed an increase (E_post, 7.5%; E_pre, 6.4%) following kefir consumption.

These findings, supported by both genus- and species-level analyses, show alterations in microbial diversity and composition. Further details on the species-level changes are available in the Appendix A.

### 3.4. Association Between Gut Microbiota Composition and Athletic Performance in Kefir-Consuming Athletes

The relationship between gut microbiota composition and finishing speed performance was investigated in kefir-consuming athletes categorized into the low-, medium-, and high-performance groups. At the species-level, distinct microbial signatures were observed across the performance categories (Table 2).

The low-performance group was characterized by a higher relative abundance of *Dorea formicigenerans* (*D. formicigenerans*; 28.43%) and *Oxalobacter formigenes* (*O. formigenes*; 25.89%); however, these species were less prominent in the medium- and high-performance groups. The medium-performance group displayed an enrichment of species, including *Coprococcus catus* (*C. catus*; 31.00%) and *A. muciniphila* (12.67%), whereas the high-performance group exhibited a higher abundance of *Faecalibacterium prausnitzii* (*F. prausnitzii;* 27.19%) and *P. copri* (17.48%).

The relative abundances of key microbial species across VO_2_max performance levels (low, medium, and high) in kefir-consuming athletes are summarized in Table 3. *F. prausnitzii* was consistently dominant across all performance levels, with the highest abundance observed in the medium-performance group (31.00%), followed by a slight decrease in the high-performance group (27.18%). Similarly, the relative abundance of *P. copri* was highest in the low-performance group (25.89%) and decreased at higher performance levels, reaching 17.47% in the high-performance group. Conversely, the medium-performance group demonstrated a marked increase in *A. muciniphila* (12.67%) and *Ruminococcus bromii* (*R. bromii*; 9.52%).

### 3.5. Interplay Between Gut Microbiota, Dietary Intake, and Athletic Performance Metrics in Kefir-Consuming Athletes

This section examines the relationship between the gut microbiota profile, diet, and athletic performance measures, including VO_2_max and finishing speed among the kefir-consuming athletes. The heatmap (Figure 6) highlights the significant positive and negative correlations.

#### 3.5.1. Gut Microbiota and Athletic Performance Correlations

Finishing speed (km/h)

There was a moderate positive correlation between *F. prausnitzii* and finishing speed (r = 0.34, *p* = 0.13) and between *Blautia* and finishing speed (r = 0.38, *p* = 0.09); however, these correlations were not significant.

VO_2_max (mL.kg^−1^.min^−1^)

There was a moderate positive correlation between *F. prausnitzii* and VO_2_max (r = 0.32, *p* = 0.16) and between *Blautia* and VO_2_max (r = 0.38, *p* = 0.09); however, these correlations were not statistically significant. Similarly, a moderate negative correlation was observed between *Bacteroides* and VO_2_max (r = −0.31, *p* = 0.17), but this association was also not statistically significant.

#### 3.5.2. Gut Microbiota and Body Composition Correlations

Fat Mass (kg)

A moderate negative correlation was observed between *Veillonella* and fat mass (r = −0.49, *p* = 0.02). While *P. copri* also showed a moderate negative correlation with fat mass (r = −0.31, *p* = 0.17), this association was not statistically significant. Similarly, *Bifidobacterium* exhibited a moderate positive correlation with fat mass (r = 0.35, *p* = 0.12), but this relationship was not statistically significant.

Fat-Free Mass (kg)

A moderate positive correlation was observed between fat-free mass and *R. faecis* (r = 0.45, *p* = 0.04). While *R. faecis* also showed a moderate positive correlation with body fat percentage (r = 0.33, *p* = 0.14), this association was not statistically significant. Conversely, moderate negative correlations were observed with *Bacteroides* (r = −0.47, *p* = 0.03) and *Oscillospira* (r = −0.53, *p* = 0.01).

Body Fat (%)

Moderate negative correlations were observed between body fat percentage and *Prevotella* (r = −0.48, *p* = 0.03) and *Veillonella* (r = −0.53, *p* = 0.01). Although a moderate positive correlation was observed between body fat percentage and *R. faecis* (r = 0.33, *p* = 0.14), this association was not statistically significant.

#### 3.5.3. Gut Microbiota and Dietary Intake Correlations

Dietary Fiber Intake (g)

Although a moderate positive correlation was observed between dietary fiber intake and *F. prausnitzii* (r = 0.38, *p* = 0.09), this association was not statistically significant. In contrast, a moderate positive correlation was observed between dietary fiber intake and *Oscillospira* (r = 0.45, *p* = 0.04).

Protein Intake (g, %)

Protein intake was significantly correlated with various microbial populations in the study. Total protein intake (g) showed a strong positive correlation with *Oscillospira* (r = 0.55, *p* = 0.01). Furthermore, dietary protein intake percentage showed strong positive correlations with *Lactobacillus helveticus* (r = 0.71, *p* < 0.01) and *Blautia* (r = 0.76, *p* < 0.001). Additionally, vegetable protein intake (g) showed a moderate positive correlation with *Oscillospira* (r = 0.51, *p* = 0.018).

Fat Intake (g)

A strong positive correlation was noted between fat intake and *Oscillospira* (r = 0.71, ***p*** < 0.01).

## 4. Discussion

Dietary interventions in modulating the gut microbiota and their effects on athletic performance have garnered increasing research attention owing to the increasing focus on the bidirectional relationship between nutrition and physical activity. Although several studies have been conducted to investigate the effects of dietary patterns or certain nutrients on the gut microbiota and its functions, research on targeted interventions, particularly among professional athletes, remains lacking. This study contributes to the literature by investigating the effects of regular kefir consumption on gut microbial diversity, body composition, and performance metrics in professional female soccer players. By integrating microbiome studies with performance and physio-anthropometric variables, the findings of this study provide a novel insight into the positive effects of incorporating functional food in the diet of athletes to enhance performance and well-being.

### 4.1. Dietary Intake as a Gut Microbiota Modulator

Dietary intake is considered to significantly affect gut microbiota composition, diversity, and activity. The gut microbiota is highly dynamic, and it has been observed that it can significantly change within 1 day of altering the diet [21]. The growth and activity of microorganisms are influenced by macronutrients and certain dietary patterns in a specific manner. For instance, diets high in complex carbohydrates and dietary fiber are associated with increased microbial richness and diversity, whereas high-protein or high fat diets can negatively affect the balance of the gut microbiota by encouraging the growth of proteolytic or bile-tolerant bacteria [21,22,23,24]. The influence of dietary components on the gut microbiota transcends microbial composition modulation as it affects metabolic pathways for energy and thus the host health, indicating that it is crucial to comprehend these interactions to design effective dietary strategies that enhance gut health and subsequently athletic performance.

Carbohydrates are widely recognized to significantly affect the gut microbiota to a large extent. Dietary fiber consumption can therefore support the growth of SCFA-producing bacteria, *Roseburia* and *Faecalibacterium*, which are essential in maintaining the integrity of the gut and reducing inflammation, respectively [25,26]. In athletes, a higher dietary fiber intake has been shown to enhance the abundance of *Prevotella*, a bacterium that has been linked with improved energy utilization and performance [27,28]. Conversely, fiber-deficient diets can restrict the diversity and stability of the microbiome and may therefore affect the gut health during physical stress. Our findings further support this relationship as dietary fiber intake showed a moderate positive correlation with the abundance of *F. prausnitzii* (r = 0.38) and *Oscillospira* (r = 0.45), which are both associated with a healthy gut and efficient metabolism. These findings suggest that dietary fiber may play a key role in improving microbial diversity and positively influencing the gut microbiota profile of athletes.

Proteins essential for muscle repair and recovery can negatively affect the gut microbiota. High-protein diets have been shown to enhance the growth of proteolytic bacteria, which results in the formation of toxic compounds, including ammonia and sulfides [29,30]. However, the source of protein seems to have different effects; plant-based proteins are more likely to positively influence the microbial population, whereas a high intake of animal-based proteins may cause an imbalance in the gut microbiota [31,32]. Moreover, our study revealed a strong positive correlation between total protein intake and abundance of *Oscillospira* (r = 0.55) and a moderate positive correlation between vegetable protein intake and *Oscillospira* (r = 0.51). *Oscillospira* is known for its ability to produce SCFAs, including butyrate, and is believed to contribute to athletic performance [33]. These findings highlight the potential of protein sources, particularly plant-based proteins, in shaping beneficial microbial populations and promoting gut health in athletes.

Furthermore, fats can significantly affect the gut microbiota composition, especially with respect to their type and quantity. In particular, omega-3 polyunsaturated fatty acid-rich diets have been found to positively correlate with the abundance of *Bifidobacteria* and *Akkermansia*, which have anti-inflammatory effects and are involved in SCFA production [34]. In contrast, saturated fat-rich diets can reduce microbial diversity and promote the growth of bile-tolerant bacteria, including *Bacteroides* [35]. Our findings provide additional nuance to this relationship as fat intake showed a strong positive correlation with *Oscillospira* abundance (r = 0.71), a genus potentially associated with metabolic regulation. This strong association suggests that total fat intake may play a role in promoting beneficial microbial populations, although the specific effects of different types of fats remain unclear. However, as our study did not focus on examining the effects of different types of fat on the microbiota, our current findings only show the effects of total fat intake on the gut microbiota. Future research should differentiate the roles of saturated, unsaturated, and omega-3 fatty acids in modulating gut microbiota composition to provide a more comprehensive understanding of their individual effects.

### 4.2. Gut Microbiota and Body Composition

The gut microbiota significantly impacts host metabolism regulation, affecting key aspects of body composition, including fat mass, fat-free mass, and overall body weight. Some microbial communities have been associated with metabolic processes that regulate energy deposition, lipolysis, inflammation and therefore body composition. For instance, bacteria, including *Roseburia* and *Faecalibacterium*, are associated with the production of SCFAs, which are essential in energy metabolism and anti-inflammatory processes, whereas some proteolytic or bile-tolerant bacteria may cause obesity or dysbiosis [36,37,38]. Furthermore, the interdependence between the microbial community and diet along with physical activity demonstrates that the ratio of beneficial and potentially pathogenic microorganisms is not fixed but can be influenced by certain factors. This information explains how variations in the gut microbiota can be applied to control the body composition in athletes. In our study, a moderate positive correlation was observed between fat-free mass and the abundance of *R. faecis* (r = 0.45), a genus known for its ability to produce SCFAs that are crucial for energy metabolism and combating inflammation. This finding is consistent with that of a previous study that suggested that SCFA-producing bacteria play a vital role in maintaining lean body mass and regulating energy metabolism [36,37]. These results highlight the potential of modulating gut microbiota composition as a strategy to optimize body composition and athletic performance.

Conversely, fat mass was moderately negatively correlated with *P. copri* (r = −0.31) and *Veillonella* (r = −0.49), two genera of bacteria associated with increased carbohydrate metabolism and lactate utilization. This finding is consistent with that of an earlier study that highlighted that the presence of carbohydrate-fermenting bacteria can help improve metabolic efficiency and reduce adiposity in athletes [27,28]. In contrast, *Bifidobacterium* was moderately positively correlated with fat mass (r = 0.35), suggesting its involvement in energy deposition or utilization. These results suggest that microbial populations can influence fat metabolism differently, with certain genera, such as *P. copri* and *Veillonella*, promoting metabolic efficiency, while others, like *Bifidobacterium*, may contribute to energy storage. This finding implies that some of the bacteria that are deemed beneficial for gut health may have context-dependent effects on fat metabolism, influencing it both positively and negatively depending on the metabolic state [37].

In a similar manner, other parameters including body fat percentage, which is a measure of total adiposity, were well correlated with the microbial populations. Moderate negative correlations were observed with *Prevotella* (r = −0.48) and *Veillonella* (r = −0.53), aligning with the current understanding that these bacteria are beneficial in fat metabolism and are therefore anticipated to be more abundant in lean individuals. Interestingly, *R. faecis* was moderately positively correlated with body fat percentage (r = 0.33), suggesting that the function of this bacterium in energy metabolism could be context-dependent and influenced by diet and exercise [34,39]. These findings highlight the intricate interplay between the gut microbiota and anthropometric measures, wherein microbial communities can exert both beneficial and potentially adverse impacts on fat metabolism depending on environmental and physiological factors.

Dietary intake was also observed to play a significant role in these interactions, wherein it alters the microbial profile that in turn regulates body composition. Higher dietary fiber intake, which was observed in our study, showed a moderate positive correlation with the abundance of *F. prausnitzii* (r = 0.38) and *Oscillospira* (r = 0.45), both of which are associated with enhancing fat-free mass and reducing inflammation. Similarly, protein intake showed a strong positive correlation with *Oscillospira* abundance (r = 0.55), highlighting its beneficial effects in promoting lean body mass and mitigating the adverse effects of high-protein intake [29,31]. These findings underscore the critical role of dietary patterns in shaping microbial populations that influence body composition and metabolic health.

These findings support the hypothesis that the gut microbiome contributes to the modulation of metabolism and inflammation, thereby influencing body composition. The moderate and strong correlations observed in our study suggest that dietary interventions targeting the stimulation of SCFA-producing bacteria, such as *F. prausnitzii* and *Oscillospira*, while reducing proteolytic or bile-tolerant bacteria, could offer novel approaches for optimizing body composition in athletes. Further investigations should examine these relationships in greater depth, particularly focusing on the combined effects of diet and exercise on anthropometric and metabolic outcomes. Future studies with larger sample sizes and longer intervention durations are essential to validate these preliminary findings and explore potential causal mechanisms.

### 4.3. Role of the Gut Microbiota in Athletic Performance

The relationship between the gut microbiota and athletic performance has been recognized as a modulator of energy metabolism, inflammation, and recovery. In our study, the gut microbiota composition of athletes classified by performance levels (low, medium, and high) based on the IFT results demonstrated varying microbial profiles.

At the genus and species levels, the high-performance group had a higher abundance of *F. prausnitzii* (27.19%) and *P. copri* (17.48%), both of which are associated with improved energy utilization and endurance performance. *F. prausnitzii* produces SCFAs and is widely recognized for its positive effects on the gut and its ability to reduce inflammation throughout the body, thereby facilitating the efficient use of energy during high-intensity physical activities [40,41]. Similarly, *Prevotella* species have been associated with carbohydrate metabolism and energy production, which are highly beneficial for athletes who engage in endurance exercises [27,28]. Conversely, the low-performance group showed a greater abundance of *D. formicigenerans* (28.43%) and *O. formigenes* (25.89%), genera that are not frequently associated with efficient energy utilization by metabolic processes. Furthermore, our findings show the medium-performance group as a potential transitional profile, which is also populated by *A. muciniphila* in significant quantities (12.67%) along with *C. catus* (31.00%). The presence of A. muciniphila in this group suggests its role in regulating gut mucosal integrity and metabolism, which has been linked to enhanced exercise capacity and decreased adiposity [34]. Similarly, the increase in *C. catus* in this group highlights its potential significance in maintaining gut health and energy metabolism during moderate-intensity physical activities. These results support the hypothesis that microbial profile alterations are closely linked with varying physical performance levels, suggesting adaptability in the gut microbiota to meet the energy demands of exercise.

Besides the differences in the microbiome composition, microbial diversity was also revealed as a significant factor. Higher microbial diversity as measured by the Shannon and Chao1 indices was moderately positively correlated with VO_2_max and finishing speed in the high-performance group. This finding agrees with those of previous studies that reported that increased microbial diversity and taxon richness were positively correlated with enhanced metabolic health and exercise capacity [42,43]. These results suggest that microbial diversity may play a critical role in optimizing physical performance by supporting metabolic flexibility and resilience during exercise.

More specifically, the findings of this study also support those of previous studies by showing that *F. prausnitzii* was moderately positively correlated with both VO_2_max (r = 0.32) and finishing speed (r = 0.34), which strengthens the argument regarding the role of this bacteria in supporting aerobic capacity. Conversely, *Bacteroides* was moderately negatively correlated with these performance measures (VO_2_max, r = −0.31; finishing speed, r = −0.33), suggesting that these bacteria can be a marker of decreased metabolic efficiency. These results highlight the complex interplay between specific microbial taxa and athletic performance, where certain bacteria may enhance metabolic outcomes while others may hinder them.

SCFAs, the major products of the gut microbiota, play a significant role in linking microbial composition to athletic performance. Propionate and butyrate, which are SCFAs, are used by the liver and muscle cells as energy fuels during exercise, thereby improving endurance and delaying exhaustion [36,40]. For example, *Veillonella*, a genus enriched in athletes, metabolizes exercise-induced lactate into propionate, which can be utilized as an alternative energy source [44]. In our study, SCFA levels were not directly quantified, which is indirect evidence suggesting that the enrichment of SCFA-producing bacteria, including *F. prausnitzii*, *R. faecis*, and *A. muciniphila*, in higher-performing athletes can positively impact energy metabolism and exercise performance. Future studies should aim to directly measure SCFA levels to establish a clearer link between microbial metabolites and athletic performance.

Furthermore, the results of the present study support those of previous studies, indicating that microbial communities can regulate inflammation and oxidative stress, which are both essential in sports, training, and recovery. For example, the anti-inflammatory properties of *A. muciniphila* and *F. prausnitzii* can help in reducing muscle soreness and improving the recovery process, thereby supporting enhanced performance in endurance exercises. These findings highlight the potential of targeting anti-inflammatory microbial taxa as a strategy to optimize athletic recovery and performance. Future studies should further explore these mechanisms by integrating direct measures of inflammatory markers and oxidative stress levels.

Moreover, the abundances of *Blautia* and *R. bromii* were increased in the medium-performance group; *Blautia* is known to regulate glycogen storage, whereas *R. bromii* is involved in carbohydrate metabolism, which serves as an energy source during exercise [43].

Considering the information on performance sports, the increase in *P. copri* and *F. prausnitzii* in high-performance athletes also supports previous research that demonstrated that these organisms were involved in improved metabolic processes. Scheiman et al. reported that genes associated with lactic and propionic acid metabolism in the fecal microbiome were upregulated in marathon runners, and the isolated strains enhanced endurance by 13% in preclinical studies [44]. Even if these results require further validation in human models, these data are suggestive of the fact that the gut microbiota can play a significant role in the performance.

Therefore, the differences in the microbial profiles that were identified in our study among the participants with different performance levels indicate the possibility of using the gut microbiota as a biomarker and a modifiable factor for athletic performance. Furthermore, the differences in the ratios of SCFA-producing bacteria and anti-inflammatory bacteria were higher among higher-performing athletes, indicating that they contribute to energy metabolism, reduce inflammation, and enhance recovery. Further research should also involve longitudinal analyses to establish the cause–effect relationship between the gut microbiome and athletic performance in addition to the possibility of targeted dietary interventions, for instance kefir, to shape the microbial profile and subsequently the performance.

### 4.4. Limitations and Future Directions

This study had some limitations. The exclusive focus on female athletes may limit the generalizability of the findings to male athlete populations. Furthermore, as this study was conducted for a 4-week duration, which is consistent with previous research and sufficient to show noticeable shifts in the microbiota and performance, further studies should be conducted with a longer duration to monitor the adaptations that occur over time as well as the potential advantages of consuming kefir for gut health and sports performance in the long run.

Although we controlled key factors influencing the gut microbiota, such as gender, age-based randomization, and standardized training loads, baseline differences in microbial diversity between groups were observed. These differences may be attributed to uncontrolled factors such as dietary habits, lifestyle, or environmental exposures prior to the intervention. Future studies should account for these factors during participant recruitment and pre-intervention assessments to further minimize variability.

Additionally, the menstrual cycle phase of participants was not considered, which may influence athletic performance and physiological outcomes. This decision was made to prioritize consistency in training loads and maintain an adequate sample size for statistical significance. Future studies should consider incorporating menstrual cycle phases to further refine the findings.

To elucidate the mechanisms through which kefir consumption exerts its effects on metabolism and athletic performance, further research should also integrate direct assessments of SCFAs along with other microbial metabolites. Furthermore, understanding the impact of different kefir doses and dietary habits in various athletic groups will optimize the potential of dietary interventions. Future personalized nutrition approaches according to the individual microbiome may also enhance the effectiveness of such interventions and provide a systematic approach for improving the performance and recovery of athletes.

## 5. Conclusions

This study demonstrates how kefir can be used as a functional food for altering the gut microbiota and enhancing athletic performance. The participants who regularly consumed kefir had a higher microbial diversity, relative enrichment of specific microbiota, and better performance indicators. These results also underscore the need to integrate nutritional measures within the training schedules of athletes to enhance their performance and well-being. To determine the best dietary interventions that modulate the microbiota for enhanced performance in athletes, further studies are warranted.

## Figures and Tables

**Figure 1 nutrients-17-00512-f001:**
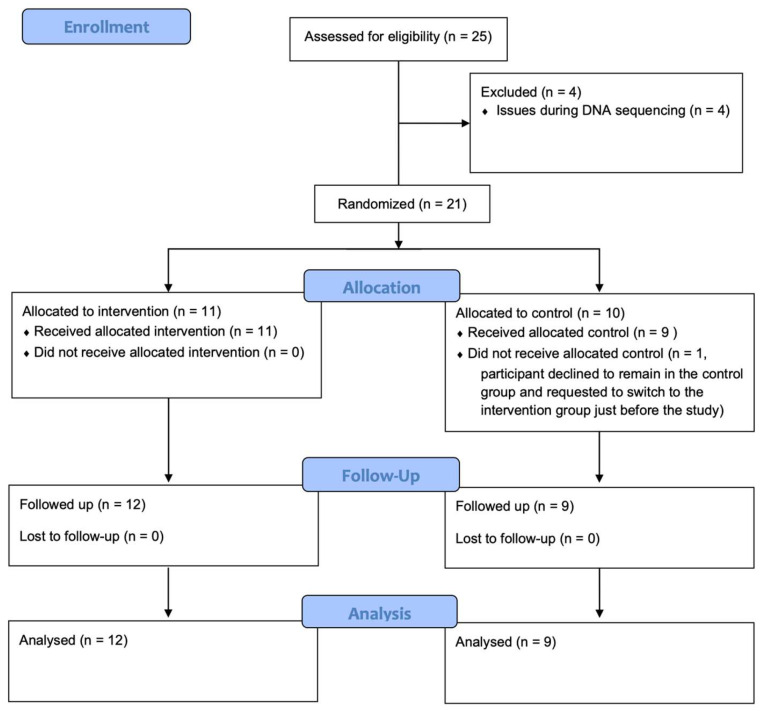
Participant flow diagram.

**Figure 2 nutrients-17-00512-f002:**
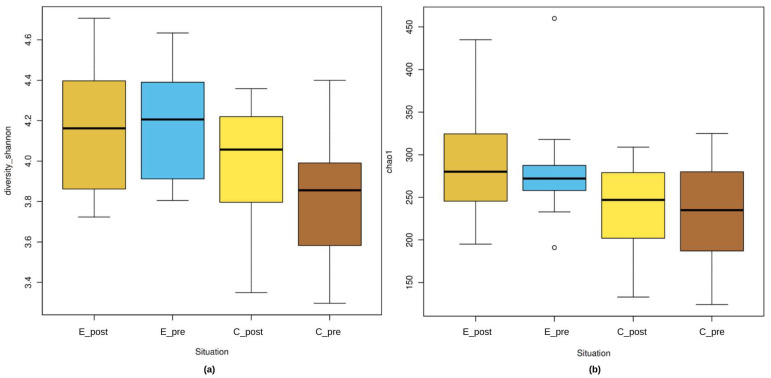
Shannon diversity and Chao1 richness indices for the experimental and control groups. The Shannon diversity index (**a**) and Chao1 richness index (**b**) are used for evaluating gut microbial diversity and richness across the experimental (E_post, E_pre) and control (C_post, C_pre) groups. The E_post group showed slightly higher microbial diversity and richness compared to E_pre, but the difference was not statistically significant (*p* > 0.05).

**Figure 3 nutrients-17-00512-f003:**
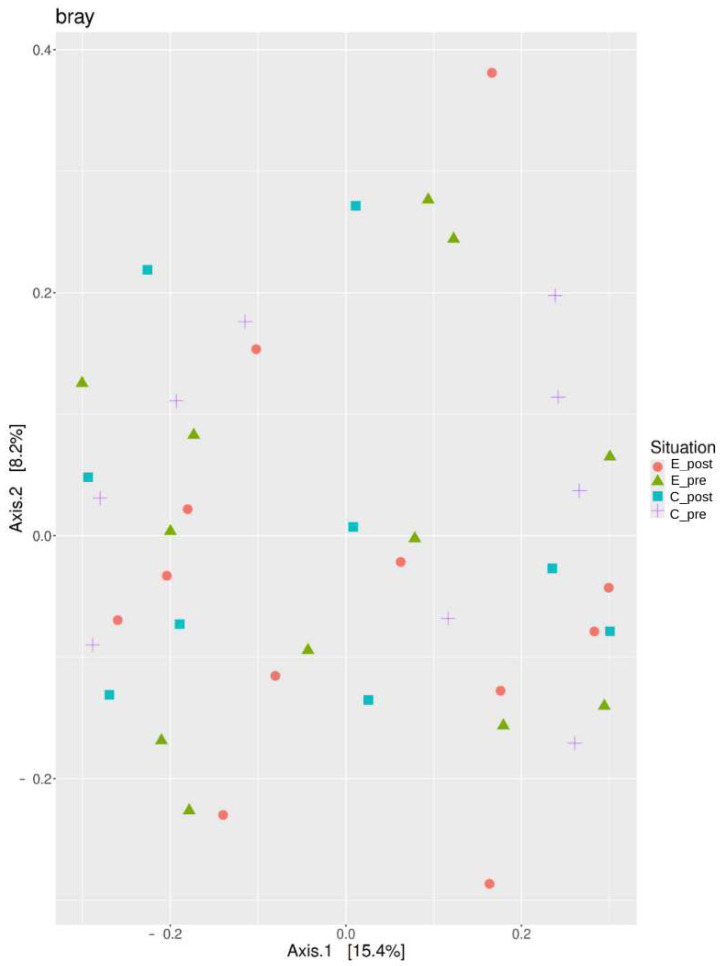
Principal coordinate analysis (PCoA) based on Bray–Curtis distances. This PCoA plot illustrates the microbial compositional differences between the experimental (E_post, E_pre) and control (C_post, C_pre) groups based on Bray–Curtis distances. The ellipses represent group clustering, with the E_post and E_pre groups showing closer proximity, indicating compositional similarity. In contrast, the C_post and C_pre groups exhibit more dispersed clustering, suggesting greater variability in the microbial composition.

**Figure 4 nutrients-17-00512-f004:**
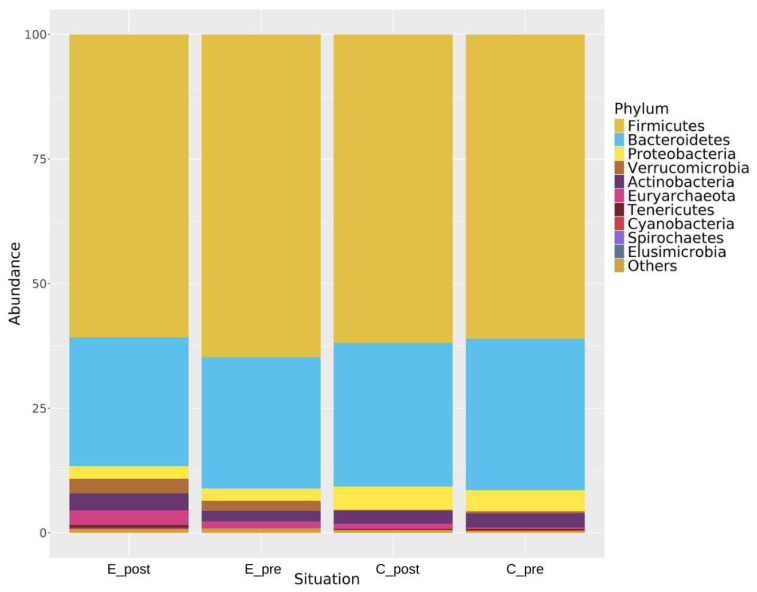
Relative abundance of bacterial phyla across different conditions (E_post, post-experimental; E_pre, pre-experimental; C_post, post-control; C_pre, pre-control). The dominant phyla, *Firmicutes* and *Bacteroidetes*, are highlighted, along with changes in minor phyla, such as *Verrucomicrobia* and *Euryarchaeota*.

**Figure 5 nutrients-17-00512-f005:**
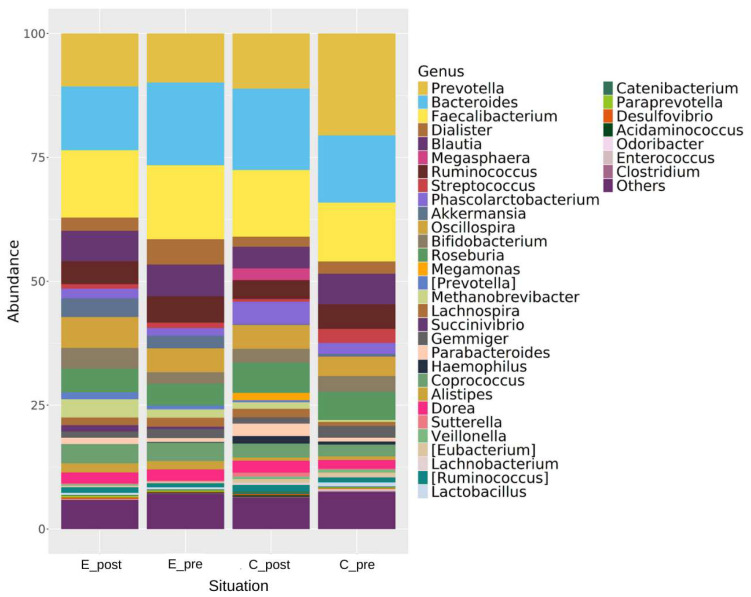
Relative abundance of bacterial genera across different conditions (E_post, post-experimental; E_pre, pre-experimental; C_post, post-control; C_pre, pre-control). Dominant genera, including *Prevotella*, *Bacteroides*, and *Faecalibacterium*, are highlighted alongside changes in other genera, including *Akkermansia* and *Bifidobacterium*.

**Figure 6 nutrients-17-00512-f006:**
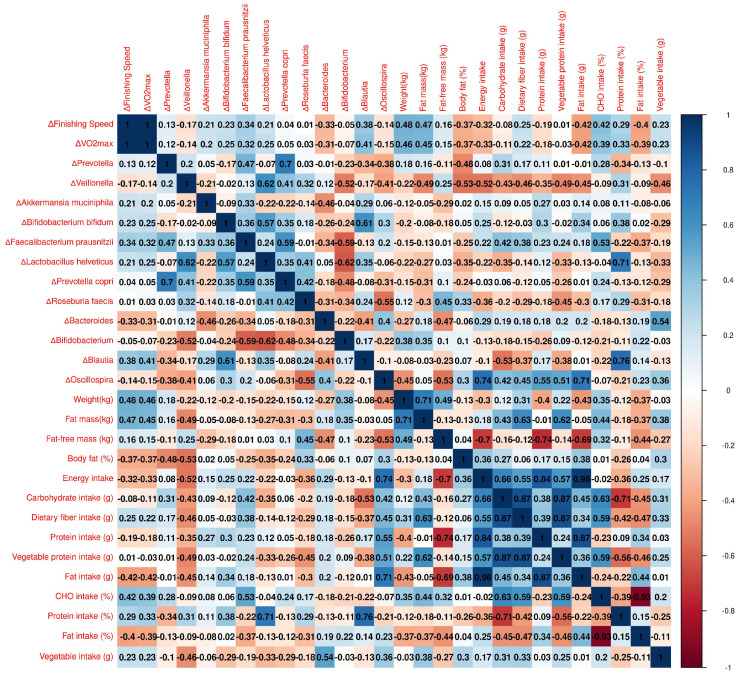
Correlations between dietary, microbiota, and performance metrics in kefir-consuming athletes. Heatmap representation of correlations. The heatmap illustrates the Spearman correlation coefficients among microbial taxa, body composition parameters, and dietary intake variables. Positive correlations are indicated by blue tones, whereas negative correlations are shown in red tones, with the intensity of the color reflecting the strength of the correlation. This visual representation provides an overview of the interactions between microbial populations, dietary components, and performance-related metrics.

**Table 1 nutrients-17-00512-t001:** Baseline characteristics of the participants.

Variable	Experimental (Mean ± SD)	Control (Mean ± SD)
Age (years)	24.42 ± 2.52	22.14 ± 3.61
Weight (kg)	60.32 ± 5.44	58.03 ± 7.64
Fat mass (kg)	10.58 ± 2.37	10.63 ± 3.43
Fat-free mass (kg)	48.16 ± 3.58	46.88 ± 5.49
Body fat (%)	17.02 ± 2.49	16.43 ± 2.80

SD, standard deviation.

**Table 2 nutrients-17-00512-t002:** Key microbial species and their relative abundances across finishing speed performance levels (low, medium, and high) in kefir-consuming athletes.

Species	Low (%)	Medium (%)	High (%)
*Faecalibacterium prausnitzii*	-	0.73	27.19
*Prevotella copri*	7.87	-	17.48
*Akkermansia muciniphila*	0.83	12.67	6.12
*Dorea formicigenerans*	28.43	-	-
*Oxalobacter formigenes*	25.89	-	-
*Coprococcus catus*	0.52	31.00	0.38

**Table 3 nutrients-17-00512-t003:** Key microbial species and their relative abundances across VO_2_max performance levels (low, medium, and high) in kefir-consuming athletes.

Species	Low (%)	Medium (%)	High (%)
*Faecalibacterium prausnitzii*	28.43	31.00	27.18
*Prevotella copri*	25.89	5.36	17.47
*Roseburia faecis*	7.86	6.62	6.36
*Ruminococcus bromii*	5.37	9.52	3.21
*Akkermansia muciniphila*	0.82	12.67	6.11
*Gemmiger formicilis*	2.65	4.22	4.17

## Data Availability

The original contributions presented in the study are included in the article, and further inquiries can be directed to the corresponding author.

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
