# Peer review of "Effects of Kefir Consumption on Gut Microbiota and Athletic Performance in Professional Female Soccer Players: A Randomized Controlled Trial"

_nutrients, 2025, doi:10.3390/nu17030512_

Round 1

Reviewer 1 Report

Comments and Suggestions for Authors

Introduction

The authors failed to claim the hypotheses of the study, which are the bases of the experimental research. This lack limits the proper interpretation of results. 

Methods section

Adequate reference for the  30-15 Intermittent Fitness Test (IFT) should be reported.

Considering the study design, statistical analysis at baseline should be reported. Differences at baseline could influence the overload findings.

Authors did not considered the menstrual cycle phase of participants, even if with self reported information.

Provide a standard definition of the 2 groups, experimental and control, since different terminologies have beed used across the different sections of the manuscript. 

A clearer description of statistical analysis should be reported. "Permutational multivariate analysis of variance (PER-MANOVA)" is not stated in this section but reported in results section.

Results seciton

Results section does not provide a clear understanding of the findings. Moreover, interpretation of results should not be reported in this section but in discussion section. 

A mismatch exist between figures order in text and caption. Specifically figure 1 and/or 2. 

Descriptive data (MEAN MEDIAN SD) are not provided for variables. Similarly statistical values (i.e., p values) are not consistently and precisely reported. Section 3.2.1. lacks for those values. Moreover, in section 3.2.1. it is stated that "K_post group had the highest microbial diversity and richness", which does not clearly emerged compared to K_pre group from figure. Moreover, it seems that diversity is higher in the K group, both at pre and post, compared to N group. How do authors explain the difference at pre between the 2 groups?

Authors affirm the following:

"However, when compared using statistical tests, no difference was noted between K_post and K_pre (p > 0.05). The distinction observed between K_post and the control groups N_post and N_pre suggest that kefir can influence the levels of gut microbial richness"

This results cannot prove any effect of Kefir. Moreover it seems difference at baseline might influence the over results. 

 What does MRPP stand for? 

Section 3.4. Analysis of associations is not reported in statistical analysis. Data presented in table 2 and 3 are not related to association analysis. 

Section 3.5.1. Analysis of associations is not reported in statistical analysis. Ranges for coefficients of correlation should be reported in statistical to address the interpretation of coefficient. As a matter, authors declared associations between variables for r = 0.34 and  r = 0.38 and others. Regardless the p values, those coefficients should be better interpreted.

Discussion sections 

The evaluation of discussion section is limited by the lack of clarity in results interpretation and reporting.

I believe the analysis of correlations is not cable to support the findings.

Author Response

Dear Sir/Madam,

We sincerely thank you for your constructive and detailed review comments on our manuscript. Your valuable recommendations and insights have greatly contributed to improving the quality of our work. All revisions made to the manuscript are highlighted in yellow for your convenience.

We look forward to your feedback and remain at your disposal for any further queries.

Sincerely,

Ece ÖneÅŸ

Acıbadem University, Institute of Health Sciences

Comment-1:

The authors failed to claim the hypotheses of the study, which are the bases of the experimental research. This lack limits the proper interpretation of results. 

Response-1:

Thank you very much for your valuable feedback. We have addressed your comment by clearly stating the hypotheses in the revised manuscript, which are now highlighted in yellow at lines 66–68. Your suggestion has helped us improve the clarity and scientific foundation of our study, and we sincerely appreciate your contribution.

Comment-2:

Adequate reference for the  30-15 Intermittent Fitness Test (IFT) should be reported.

Response-2:

Thank you for your valuable feedback. We have addressed your comment by adding a new reference (Reference 10) to support the reliability of the 30-15 Intermittent Fitness Test as a valid and reliable measure for use in field settings and sports research. This addition has been highlighted in yellow at lines 152–154, along with the corresponding reference in the reference list. Additionally, we emphasized the validity of this test specifically for female soccer players, as demonstrated in the study already cited in the manuscript.

Comment-3:

Considering the study design, statistical analysis at baseline should be reported.

Differences at baseline could influence the overload findings.

Response-3:

Thank you for your insightful comment regarding the baseline characteristics and their potential influence on the findings. To address this, we performed statistical analyses to confirm the similarity of baseline characteristics between the intervention and control groups. As shown in Table 1, the groups were comparable in terms of age, weight, fat mass, fat-free mass, and body fat percentage. Statistical analysis confirmed no significant differences between the groups at baseline (p > 0.05 for all comparisons), as highlighted in yellow at lines 222-223.

For your reference, the results of these analyses are as follows:

  • Age (years): t = 0.518, p = 0.661
  • Weight (kg): t = 0.244, p = 0.832
  • Fat Mass (kg): t = -0.012, p = 0.992
  • Fat-Free Mass (kg): t = 0.195, p = 0.866
  • Body Fat (%): t = 0.157, p = 0.890

These findings support the robustness of our study design by confirming the comparability of the groups at baseline. Thank you for helping us improve the clarity and methodological rigor of our manuscript.

Comment-4:

Authors did not considered the menstrual cycle phase of participants, even if with self reported information.

Response-4:

Thank you for your valuable comment regarding the consideration of the menstrual cycle phase. In our study, training load was considered a more critical confounding factor than the menstrual cycle phase. To control for this, we included players from the same team to ensure consistency in training conditions.

While we acknowledge the potential influence of the menstrual cycle on the study outcomes, excluding participants based on menstrual cycle phases would have made it challenging to achieve statistical significance due to the reduced sample size. Therefore, we opted to collect data without accounting for menstrual cycle phases. However, following your insightful suggestion, we have added this point as a limitation in the revised manuscript. This addition has been highlighted in yellow at lines 659–663.

We greatly appreciate your feedback, which has helped us improve the clarity and transparency of our study.

Comment-5:

Provide a standard definition of the 2 groups, experimental and control, since different terminologies have beed used across the different sections of the manuscript. 

Response-5:

Thank you for your valuable comment. We have carefully reviewed the manuscript and standardized the terminology for the two groups as experimental group and control group throughout the text. Additionally, all figures have been updated to reflect this consistent terminology.

To ensure clarity, all changes have been highlighted in yellow in the revised manuscript. Your suggestion has greatly improved the coherence and readability of our study, and we sincerely appreciate your feedback.

Comment-6:

A clearer description of statistical analysis should be reported. "Permutational multivariate analysis of variance (PER-MANOVA)" is not stated in this section but reported in results section.

Response-6:

Thank you for your valuable comment regarding the clarity of the statistical analysis section. We have revised the "Materials and Methods" section to include a clearer description of the statistical analyses performed. Specifically, we have now explicitly stated the use of permutational multivariate analysis of variance (PERMANOVA) to analyze microbial compositional differences between groups. This description has been added to the revised manuscript at lines 195–199, highlighted in yellow.

We greatly appreciate your feedback, which has helped us improve the clarity and completeness of our methods section.

Comment-7:

Results section does not provide a clear understanding of the findings. Moreover, interpretation of results should not be reported in this section but in discussion section. 

Response-7:

Thank you for your valuable comment regarding the inclusion of interpretations in the Results section. We have carefully reviewed the Results section and removed all interpretative statements, ensuring that only objective findings are reported. Any comments or interpretations have been relocated to the Discussion section, where they are more appropriate.

Please note that these revisions were directly made in the text without highlighting to maintain the manuscript's readability. We sincerely appreciate your feedback, which has helped improve the clarity and structure of our manuscript.

Comment-8:

A mismatch exist between figures order in text and caption. Specifically figure 1 and/or 2. 

Response-8:

Thank you for pointing out the mismatch between the figures' order in the text and their captions. We have carefully reviewed the manuscript and corrected the discrepancy. Specifically, the reference at line 247 has been updated to "Figure 2" to ensure consistency with the figure's caption.

We appreciate your attention to detail, which has helped us improve the clarity and accuracy of our manuscript.

Comment-9:

Descriptive data (MEAN MEDIAN SD) are not provided for variables. Similarly statistical values (i.e., p values) are not consistently and precisely reported. Section 3.2.1. lacks for those values.

Response-9:

Thank you for your suggestion regarding the calculation of descriptive statistics (MEAN, MEDIAN, SD) for diversity metrics. While these calculations are not commonly included in microbiota studies, we have performed them for your review and summarized them in the attached table.

We hope this additional information is helpful for evaluating our manuscript. Please let us know if further clarification is needed.

Comment-10:

Moreover, in section 3.2.1. it is stated that "K_post group had the highest microbial diversity and richness", which does not clearly emerged compared to K_pre group from figure.

Response-10:

Thank you for pointing out the inconsistency between the text and the figure in Section 3.2.1. We have revised the statement to better reflect the findings presented in the figure. The updated text now states: "E_post group showed slightly higher microbial diversity and richness compared to E_pre, but the difference was not statistically significant (p>0.05)."

We sincerely appreciate your feedback, which has helped us improve the clarity and accuracy of our manuscript.

Comment-11:

Moreover, it seems that diversity is higher in the K group, both at pre and post, compared to N group. How do authors explain the difference at pre between the 2 groups?

Response-11:

Thank you for your valuable comment regarding the baseline differences in microbial diversity between groups. We have addressed this point by revising the manuscript to include it as a limitation. Specifically, we have noted in lines 653–658 that, despite controlling for key factors such as gender, age-based randomization, and standardized training loads, baseline differences may still arise due to uncontrolled factors like dietary habits, lifestyle, or environmental exposures. This acknowledgment highlights the complexity of microbiota research and underscores the importance of considering such variables in future studies.

We sincerely appreciate your feedback, which has helped improve the transparency and robustness of our study.

Comment-12:

Authors affirm the following:

"However, when compared using statistical tests, no difference was noted between K_post and K_pre (p > 0.05). The distinction observed between K_post and the control groups N_post and N_pre suggest that kefir can influence the levels of gut microbial richness"

This results cannot prove any effect of Kefir. Moreover it seems difference at baseline might influence the over results. 

Response-12:

Thank you for your comment regarding the interpretation of results. We acknowledge that the observed baseline differences in microbial richness between the groups could potentially influence the overall findings. In response, we have clarified in the revised manuscript that while the statistical analysis showed no significant difference between K_post and K_pre (p > 0.05), the trends observed in microbial richness and diversity are indicative rather than conclusive.

Additionally, we have included the baseline differences as a limitation in lines 653–658, noting that factors such as dietary habits, lifestyle, and environmental exposures, which were not controlled prior to the intervention, could have contributed to this variability. We emphasize that the findings should be interpreted cautiously and within the context of these limitations.

We appreciate your feedback, which has helped us refine the interpretation and presentation of our results.

Comment-13:

What does MRPP stand for? 

Response-13:

Thank you for your comment regarding the inclusion of MRPP. We have clarified the use of the Multi-Response Permutation Procedure (MRPP) in the revised manuscript. Specifically, we have added details about the application of MRPP as a complementary non-parametric method to assess microbial compositional differences between groups. This addition has been made in the Materials and Methods section between lines 199 and 203.

We appreciate your feedback, which has helped us improve the clarity and comprehensiveness of our methodology.

Comment-14:

Section 3.4. Analysis of associations is not reported in statistical analysis.

Data presented in table 2 and 3 are not related to association analysis.

Response-14:

Thank you for your comment regarding Tables 2 and 3. We would like to clarify that these tables do not present statistical association analyses. Instead, they provide descriptive data on the relative abundance of microbial species within performance-based categories (low, medium, and high performance). Species with a relative abundance of less than 1% were excluded from the tables for clarity and to focus on the most abundant taxa.

The purpose of Tables 2 and 3 is to illustrate the microbial composition trends across performance groups rather than to perform statistical comparisons. As such, no revisions have been made to these tables since they accurately reflect the intended descriptive nature of the data.

We appreciate your feedback, which has helped us clarify the purpose and scope of these tables in the context of the manuscript.

Comment-15:

Section 3.5.1. Analysis of associations is not reported in statistical analysis. Ranges for coefficients of correlation should be reported in statistical to address the interpretation of coefficient. As a matter, authors declared associations between variables for r = 0.34 and  r = 0.38 and others. Regardless the p values, those coefficients should be better interpreted.

Response-15:

Thank you for your comments regarding the correlation analysis and the interpretation of correlation coefficients. We have updated the Materials and Methods section to include details about the Spearman correlation analysis and the ranges used to interpret correlation coefficients (lines 203–206). Additionally, in the Results section, all correlation relationships have been described based on the defined correlation ranges—weak (r = 0.1–0.3), moderate (r = 0.3–0.5), or strong (r > 0.5)—regardless of the p values. These classifications have been consistently applied throughout, and all relevant changes have been highlighted in yellow for ease of review.

We appreciate your valuable feedback, which has helped us improve both the clarity and completeness of our methodological description and the interpretation of our findings.

Comment-16:

Discussion sections 

The evaluation of discussion section is limited by the lack of clarity in results interpretation and reporting. I believe the analysis of correlations is not cable to support the findings.

Response-16:

Thank you for your feedback regarding the Discussion section. Based on your suggestions, we have revised the correlation statements in both the Results and Discussion sections to provide greater clarity and context. While we acknowledge the limitations of correlation analyses, as stated in the Discussion and Limitations sections, this study serves as a preliminary exploration in an area with limited prior research. Our findings offer valuable insights and lay the groundwork for future studies that may adopt alternative methodologies to strengthen the evidence. We have emphasized throughout the manuscript that the results should be interpreted with caution and that further investigations are needed to confirm these observations.

We appreciate your thoughtful comments, which have helped us improve the quality and transparency of the Discussion section.

Reviewer 2 Report

Comments and Suggestions for Authors

The obesity pandemic is the primary global health problem. Many articles and directives have been written and produced about nutrition profiles. Fortunately, many of them pay attention to the necessity of physical activity. The last one is important on the same level as non-processed food consumption. The last several years have brought strong scientific knowledge about gut microbion and its influence on human psycho-physical health. The article entitled: Effects of Kefir Consumption on Gut Microbiota and Athletic Performance in Professional Female Soccer Players: A Randomized Controlled Trial has taken into consideration the role of microbiome condition on the results of professional sportsmen (female saucers). The idea of kefir consumption investigation and its influence on microorganism profile is interesting. Moreover, the method to answer the authors questions is correctly selected and described. The bioinformatic techniques should be described in detail. However, the trial group is too small for significant results – therefore, I recommend putting in the title preliminary studies. Also, I recommend putting the information about fat, carbohydrates, and protein kefir composition.

The article is well-written and readable, with correctly selected references. The resolution of figures should be increased. Fig. 6 is entirely unreadable.

In conclusion, this article needs further revision.

Author Response

Dear Sir/Madam,

We sincerely thank you for your constructive and detailed review comments on our manuscript. Your valuable recommendations and insights have greatly contributed to improving the quality of our work. All revisions made to the manuscript are highlighted in yellow for your convenience.

We look forward to your feedback and remain at your disposal for any further queries.

Sincerely,

Ece ÖneÅŸ

Acıbadem University, Institute of Health Sciences

Comment-1:

The bioinformatic techniques should be described in detail.

Response-1:

Thank you for your valuable comment regarding the description of bioinformatic techniques. In response, we have expanded the relevant section to provide a more detailed explanation of the bioinformatic and statistical analyses performed. These additions, including details on the pipelines, metrics, and statistical methods, have been highlighted in yellow between lines 195–203 in the revised manuscript for your review.

We appreciate your feedback, which has helped us improve the methodological clarity of the manuscript.

Comment-2:

However, the trial group is too small for significant results – therefore, I recommend putting in the title preliminary studies.

Response-2:

Thank you for your thoughtful comment regarding the sample size and the suggestion to include "preliminary study" in the title. As we have detailed the limitations of our study extensively in the manuscript, we believe it may not be necessary to add this expression to the title. However, if you consider it an essential addition, we are more than willing to revise the title accordingly.

We greatly value your insights and appreciate your guidance.

Comment-3:

Also, I recommend putting the information about fat, carbohydrates, and protein kefir composition.

Response-3:

Thank you for your valuable suggestion regarding the inclusion of kefir composition information. In response, we have added the details about the fat, carbohydrate, and protein content of the kefir used in the study. This information has been highlighted in yellow between lines 126–128 in the revised manuscript for your review.

We appreciate your insightful feedback, which has helped enhance the clarity and completeness of the manuscript.

Comment-4:

The article is well-written and readable, with correctly selected references. The resolution of figures should be increased. Fig. 6 is entirely unreadable.

Response-4:

Thank you for your positive feedback regarding the manuscript. We appreciate your comment on the figure resolutions. In response, we have increased the resolution of all figures, and the size of Figure 6 has been specifically enlarged to ensure readability. We hope this improves the clarity and visual presentation of the manuscript.

Thank you for your valuable suggestions.

Reviewer 3 Report

Comments and Suggestions for Authors

This great article examines the effect of kefir on gut microbiota, in extension coupling this to actual anthropometric measurements. I have some remarks:

Figure 2: Did you also calculate the Simpson index?

Figure 3: Although the sample size is small, could you also calculate and discuss gamma-diversity?

Figure 6: Which correlation coefficient was used? I cannot find any information on this one.

Author Response

Dear Sir/Madam,

We sincerely thank you for your constructive and detailed review comments on our manuscript. Your valuable recommendations and insights have greatly contributed to improving the quality of our work. All revisions made to the manuscript are highlighted in yellow for your convenience.

We look forward to your feedback and remain at your disposal for any further queries.

Sincerely,

Ece ÖneÅŸ

Acıbadem University, Institute of Health Sciences

Author Response:

Comment-1:

Figure 2: Did you also calculate the Simpson index?

Response-1:

Thank you for your insightful comment regarding the Simpson index. In response, we have included the results of the Simpson index analysis in the manuscript, which have been added between lines 260–262. Additionally, these findings are presented in Supplementary Materials as Figure S1 for further reference.

We appreciate your valuable feedback, which has helped enhance the completeness and clarity of our study.

Comment-2:

Figure 3: Although the sample size is small, could you also calculate and discuss gamma-diversity?

Response-2:

Thank you for your valuable suggestion regarding the calculation of gamma diversity. We have carefully considered your recommendation and discussed it with our bioinformatics expert.

Our expert has advised that performing this calculation may not yield meaningful results due to the insufficient sample size in each group (fewer than 10 participants per group). Given this limitation, we believe that the analysis would not provide reliable or interpretable outcomes. We hope this explanation clarifies our decision, and we remain open to any further suggestions or guidance you may have.

Comment-3:

Figure 6: Which correlation coefficient was used? I cannot find any information on this one.

Response-3:

Thank you for your comment regarding the correlation coefficients. In response, we have clarified the statistical methods used for the correlation analysis by adding detailed information between lines 206–209 in the revised manuscript. Specifically, we used the Spearman correlation coefficient, and the ranges for interpretation (weak, moderate, strong) were provided. Additionally, we have ensured that these ranges were consistently applied in the Results and Discussion sections, independent of p-values, to enhance clarity and coherence in reporting the findings.

We appreciate your feedback, which has helped us improve the transparency and rigor of our analysis.